# The Transactional Horizons of Greta Thunberg

Daniel Stoecklin 

Centre for Children's Rights Studies, University of Geneva (Valais Campus), 1967 Bramois, Switzerland; daniel.stoecklin@unige.ch

**Abstract:** The paper aims at developing new understandings of agency, or capacity to make a difference, which is a central issue in childhood studies. Sixteen speeches delivered by climate activist Greta Thunberg between 2018 and 2019 are analyzed. The findings reveal 5 core reflexive operations (objectification, personification, sanctification, unification and diversification) underpinning the speeches. This is conducive to the hypothesis that Greta's audience and the replications of demonstrations for climate justice are bound to 5 transactional horizons (activities, relations, values, images of self and motivations) identified as the symbolic landscapes channeling the social interactions in climate activism. Transactional horizons form a structure of intelligible categories linked to sensatory experience. These vectors of agency twist perceptual consciousness into a hierarchized reflective consciousness. The dominant perspective of agency *within* structure is challenged by this emerging paradigm of agency *through* structure, whereby the two terms are seen as fluid and sedimented states. Future directions are identified for interdisciplinary research, contributing to heightened awareness of recursive processes that may impact climate policies.

**Keywords:** agency; childhood studies; climate activism; Greta Thunberg; interdisciplinarity; reflexivity; structure; transactional horizon



## 1. Introduction

In August 2018, a 15-year-old girl sat down before the Swedish Parliament to strike for the climate instead of going to school. Greta Thunberg's action prompted one of the most important social movements in history and the biggest in terms of children's participation. On 15 March 2019, an estimated 1.6 million people in 2000 locations took to the streets, followed by many other protests. In a year, innumerable pupils went on one-day school strikes, 4 million alone on the eve of the UN Summit for Climate Action in September 2019. How could Greta Thunberg have such agency? This is the question that is addressed in this article.

Dominant accounts of agency broadly define it as the capacity to make a difference. Under this light, the agency of Greta Thunberg is impressive. In two years, she has moved from the status of an unknown girl to a prominent interlocutor in climate politics; one of the latest instances is her meeting with Angela Merkel as President of the Council of the EU, on 20 August 2020. This journey began two years before with "Fridays for Future" that have been suspended only because of the COVID-19 pandemic, probably just a temporary break in this social movement. It took a pandemic to stop the justice for climate marches. Personal attacks against Greta accusing her of being manipulated by specific lobbies and being instrumental for them did not halt her, although many other children of her age, placed in the same conditions, would feel intimidated. The pressure is tremendous because these suspicions are symptoms of a regime of truth [1] that naturalizes the neo-liberal ideology of the responsible self: someone must be responsible, and many people still doubt that teenagers like Greta are able to speak on their own. A child having political agency is an oxymoron to many, and this contributes to presenting Greta Thunberg as either a "leader" or a "puppet", a framing that is not doing justice to the millions of climate strikers

depicted as "followers" instead of social actors shaping the movement. Typically, the climate crisis re-emerges in the media almost only when Greta Thunberg says something.

The analysis of ecological movements started long ago, but the analysis of the impressive success of a single child in these movements only started recently, logically following the emergence of Greta Thunberg on the political scene, starting in 2018 with her first school strike for climate and attracting unprecedented media coverage. According to Yearley [2], Beck and Giddens have pointed out "that risks and nature worries in contemporary societies come not principally from uncontrolled natural events but from the unintended consequences of human interventions in nature" (p. 199). Two decades later, the general awareness of the adverse effects of human interventions has dramatically grown thanks notably to social movements such as climate justice and Extinction Rebellion, and to the scientific evidence of global warming that gained media attention through these movements. In 2019, Holmberg and Alvinius noted that "there is very little research on children's resistance in relation to global issues" [3] (p. 2). In 2020, still "few academic analyses have addressed the mobilization of youth in global climate politics" [4] (p. 1). Current analyses focus on the formal features of the climate social movement, the types of activism and forms of dissent [5], the type of claims and counterclaims that are made, and the use social media [3,4], or the different narratives specific to the global North/South [6]. Holmberg and Alvinius [3] conducted a thematic analysis of Greta Thunberg's speeches. They showed that the main message is resistance against the "laissez-faire" attitude prevalent in politics and identified two themes, namely (1) a need for political and social change focusing on the climate emergency, and (2) resistance targets including political leaders, capitalist ideologies, and older generations. They call the case of Greta Thunberg "abstract progressive resistance" [3] (p. 1), which, according to them, illustrates the power of children expressing themselves in this way. They consider that through the use of rapid social media, "children have managed to create opinion and equalise hierarchies between decision-makers, world leaders and the public worldwide" [3] (p. 11).

The perspective developed in this paper is also based on an analysis of Greta Thunberg's speeches, but it differs from the one done by Holmberg and Alvinius [3] both in scope and aim. The present analysis is based on 16 speeches now available, instead of the 5 published in 2019 that Holmberg and Alvinius [3] could use as the material for their analysis, and it aims at identifying the reflexive operations beneath the claims that are made. For this analysis, the concept of "transactional horizons" [7] is used. Transactional horizons are symbolic landscapes channeling social interactions. As is shown, they form a system that functions as a socio-cognitive interface allowing actors to verbally connect objects. The study case of Greta Thunberg along transactional horizons brings important developments in the understanding of agency, a central issue not only for childhood studies but for social sciences at large. It shows that agency is deployed not *within* but *through* structure. These theoretical developments, in turn, allow understanding the "starification" of Greta Thunberg as a social process that can be explained by the very "twisting movement" identified thanks to this case study: the translation from the simultaneity of sensory experience into a hierarchized mediatic discourse about this "lone girl" exemplifies the theory. Her rise on the political scene is an "interactional accomplishment" [8], echoing in her own subjective (re)constructions of reality. This is why these reconstructions of reality, expressed in her published discourses [9], are taken as the empirical material magnifying not just climate marches but more generally social dynamics. That is to say that the many replications of climate marches are indicative of the structural properties of social dynamics.

The paper is structured with the following sections: the problem statement and the theoretical framework (Section 2) are presented first, as they determine the choice of materials and methods (Section 3). The findings (Section 4) are followed by a discussion that is developed around two topics: the exercise of agency through structure (Section 5.1), and power as the naturalization of cultural pertinences (Section 5.2). The limits of the paper are presented along suggested directions for further research (Section 6). The conclusion (Section 7) finally situates the possible implications of the emerging theory on practices.

## 2. Problem Statement and Theoretical Framework

Dominant accounts of agency implicitly convey the idealized figure of the hero liberating from oppression, or at least arranging one's life despite adverse conditions. Applied to children, the narrative of the weak superseding the strong is very powerful. Nevertheless, during the last decade, a growing number of scholars in childhood studies have underlined the collective dimension and relational nature of agency [10–12]. They suggest that agency is not a property of individuals but a relational issue that can be viewed as an interactional accomplishment [8]. The case of climate activist Greta Thunberg illustrates the importance of relationality. As she is one of the most influential children (according to the UN Convention on the Rights of the Child, a child is any person under the age of 18) ever in history, the "peoplization" of politics in mass media turns her into a demiurgic icon. Avoiding this trap, sociologists unveil the "myth of the individual child" [10] by which agency is considered as a property of the self: the arrangements of reality any individual can make are always social.

Meanwhile, the "plethora of small-scale micro studies based on illustrating children's subjective active, meaning-making everyday activities" [11] (p. 129) face the growing crisis of social constructionism [13]. This probably stems from a one-sided reading of the foundational Thomasian "theorem": "if men define situations as real, they are real in their consequences" [14] (p. 571). The impact of human projections on things cannot be fully understood if one ignores the very construction of the objects that are defined as real: one should also look at how forms are extracted on a background and how their collocations become "situations". Social constructionists tend to make a kind of "social Gestaltism" as they situate the forms (Gestalt) located by individuals as a result of human institutional arrangements of the words and concepts they are currently using. They usually overlook the cognitive structures and material grounding of such arrangements. Nonetheless, power relations are not independent of cognition and material conditions. The perspectives of children, therefore, are not only concerned with relationality but also with materiality. Agency and structure, the pair of concepts transversal to childhood studies, although articulated differently in competing theories, must be re-grounded in logics that overflow empowering and capacitation approaches focusing only on human factors. While the agency–structure dichotomy has been critically assessed [12], exploring and understanding the articulations between human and non-human is the next challenge [15,16]. It is also a challenge for humanity: making sense of how subjective experience and material environment are intertwined, and acting accordingly, is required to reduce the threat of continuous global warming. Therefore, the problem that is addressed in this article is the material grounding of the interdependence between agency and structure. This problem is approached through analysis of the discourses held by climate activist Greta Thunberg.

The importance of relationality does not imply that an individual child cannot be the focus of a study: Norbert Elias developed a great analysis of configurations when he looked at another very famous child, Mozart [17]. This is a major work because he put the case in the context of the changes happening in the social status of the composer between the generation of Mozart (1756–1791) who was still viewed as an artisan, and that of Beethoven (1770–1827) who was already considered an artist. Elias underlined the interrelations between individuals and the social configurations they were experiencing, and more or less influencing. Therefore, the centration on individual children is relevant, as long as empirical demonstrations are not "restricted to examples of children making a difference to a micro relationship or set of micro relationships, with much less attention given to their impact on the wider macro generational order" [11] (p. 129).

The analytical framework used is the actor's system [18], composed of "transactional horizons" and their corresponding "modes of action" [7], as shown in Figure 1 (hereunder).

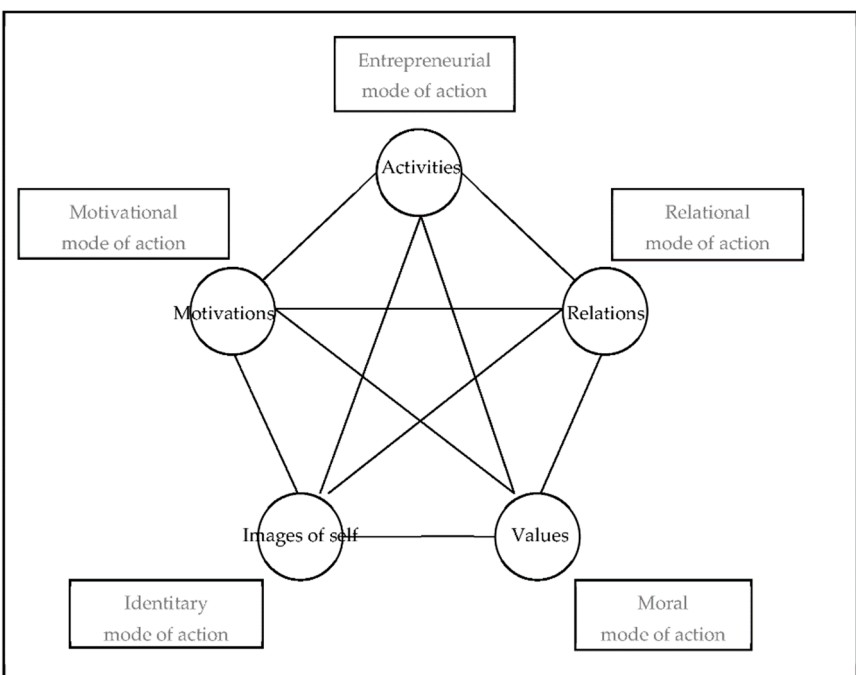

**Figure 1.** Transactional horizons and modes of action.

Transactional horizons are symbolic landscapes channeling social interactions. These symbolic landscapes are more general than "sensitizing concepts" or "directions to look at" [19] (p. 148): whereas the latter are open to being defined by the respondents, transactional horizons are not usually discussed, they are "taken for granted". The symbolic landscapes that are recurrently implied in accounts appear to be activities, relations, values, images of self and motivations: whatever the topic discussed in interviews with respondents, the items mentioned have something to do with them [18,20–24]. These symbolic landscapes harboring a diversity of experiences are taken for granted, most of the time not even mentioned, as they "go without saying". They are the "taken-for-granted highways" transporting items of experience. Mentioning them would even sound strange: asked whether working is an activity, respondents would stare at the inquirer. This question would be as "stupid" as asking whether one drives on a road. Metaphorically, transactional horizons are the implicit roads taken by social actors and on which they situate the stages of their journey.

Transactional horizons can therefore be seen as kinds of frames, but they differ from frame analysis [25] by the fact that transactional horizons are not a syntax determining human conduct, but means through which they negotiate meanings and displace the borders of inclusion and exclusion of items. The aim in the use of the notion of transactional horizons is, comparatively to Goffman's critical turn to structuralism [26], to understand agency as not determined by a structure but as being structure in a fluid state. Transactional horizons convey the idea that there is no opposition between structure and agency, but two sides of the same coin [20], a continuum between two states: structure as sedimentation of past agency, and agency as structure in the making.

Thus, whereas Goffman's frames [25] are grammatical structures underlying perception and categorization of human action, transactional horizons do not imply a turn toward structural determinism. In the present perspective, institutions are not "social structures" but instantiations of a "structure of action", conceived as "a virtual order of transformative relations" [27] (p. 17). Transactional horizons do not determine social orders: they simultaneously constrain and habilitate the social ordering of things along context-specific prioritizations. Transactional horizons are sedimenting "modes of actions" that can be defined as "typical ways of acting according to dominant thinking horizons that link together concrete items of perceived reality" [20] (p. 561). Consequently, there are

five modes of action deriving from the predominance of one transactional horizon over the others: entrepreneurial (activities), relational (relations), moral (values), identitary (images of self) and motivational (motivations).

Transactional horizons are actively used by actors to frame the course of their interaction. They are structural means for their actions, whereas constraints derive only from the constructions made with them. In sum, transactional horizons can do and undo the frames of experience [25]. A modification in the elements contained in any of these five "containers" of experience affects the whole configuration. The actor's system framework [18] was used notably to analyze accounts of children about their well-being [7,21–23]. With the intuition that children's narratives formed a system came the idea that agency itself has a structure: the view of structure within agency considers "the permanence of structural arrangements (lying) in the permanence of their representations in people's minds" [22] (p. 49). This is coherent with the constitution of the Self [28]. Hence, children's accounts reflect socialized subjectivities; they are reflections of the generalized other [28].

This solution of continuity between the organization of individual experience and the social organization [29] (pp. 23–24) leads to the perspective of "agency *through* structure" that is developed in the present paper. This opens a passage to exit the dead-end of a reified social structure [30] in the agency–structure dichotomy [12]. The link between the organization of individual experience and social organization is mediated by language: identification of factors impeding children's freely expressed views shows that language itself is a conversion factor in the child's right and capability to be heard [21]. Consequently, the structuration of ongoing subjective processes involves language, which is here considered as a translation of the sensatory experience into intelligible categories by means of transactional horizons:

"(These) symbolic landscapes which channel social interactions are framed pragmatically through standardized questions such as "what do you do?", "who do you know?", "what do you think?", "who are you?", "what do you want?" that are asked universally. These questions are currently used in social interactions because they are pragmatic tools to situate the other and hence reduce margins of error in one's own interpretations of the situation. They are pragmatic questions for inquiry (Dewey, 1938) through which actors and observers can interpret transactions. These questions in turn construct *discursive categories* like activities, relations, values, images of self, and motivations (or similar concepts) serving as common transactional horizons" [7]. It is taken for granted that anyone currently speaks about activities, relations, values, images of self and motivations. Communication breaches [25] are mostly bound to the failed indexation of what is being said, or implied by gestures, to one or several transactional horizons. This obliges the inquirer [31] to resort to these pragmatic questions. They are felt intrusive in normal conditions when discursive strategies lead to a consistent presentation of self [28]. Discursive strategies in the presentation of self can only shrink in intimate relations and in relations with children who are not yet fully mastering discursive strategies, situations where cultural artifacts are less engaged in the inquirer's reflexivity on sensatory experience.

Transactional horizons are linked together and form a system that functions as a sociocognitive interface allowing actors to verbally connect objects. Therefore, transactional horizons are the symbolic vehicles of social interaction, socially interpreted as indicators of behavior, used to situate the intentions of others. They are pragmatic means used by actors to reduce the uncertainty and potential anxiety in face of other ones' behaviors. Used to interpret behavior, these practical tools preside over the course of interactions. Transactional horizons are the *aspects* of behavior that are taken into account in Max Weber's seminal definition of social action: "Action is social insofar as its subjective meaning takes account of the behavior of others and is thereby oriented in its course" [32] (p. 4). Thus, "behavior" is apprehended through specific aspects of perpetrated acts. Moreover, behavior is more than an "act", it is a combination of acts. For Mead, "the unit of existence is the act" [33] (p. 4), and reflective consciousness is implied by the succession of acts: "Selves, minds, and our knowledge about matters of fact all emerge from acts that are experienced.

Mind cannot be separated from action. There can be no self apart from action, more specifically, from social action" (Ibid.). By using the plural form (acts), Mead [28,33] implies that an act can only make sense when related to another act, as much as a "unit" cannot stand outside of a series that forms the context or the background on which this unit is extracted. In fact, there cannot be something like an isolated social act having no connection to any other intended act. Interpretation of behavior implies extractions of forms (*Gestalt*) and connections among them. Behavior, then, is interpreted by connections made among acts (observed or supposed, visible or intended). Behavior is a combination of acts, a course or a flow of interrelated acts. Weber's analysis of action follows a methodological process whereby the motives behind acting are analyzed through their constituent parts, namely the ideal types of social action. This analytical process allows understanding behavior as combinations of these constituents. Weber identifies four types of motives:

"1. Rational action: individuals have expectations about the behaviour of others and act to take account of these expectations in order to attain their own rationally chosen outcomes.

2. Evaluative action: individuals take account of absolute values (beliefs, ethics, aesthetics or other form of behaviour) entirely for their own sake and independently of any prospects of external benefit or success.

3. Emotional actions: action based on feelings and emotions of the individual and other actors.

4. Traditional actions: actions that are based on long-established and habitually practiced traditional expectations" [34].

These are "ideal types" as they hardly exist in their pure form and as real behavior is made of their diverse combinations. Nonetheless, the combinations of (ideal typical) acts must be grounded on something different in nature from the very acts, that, so to say, bind them together. You need a weft to make a carpet. This "weft" is closer to perceptual consciousness, the ultimate basis of social processes, overlooked in sociology as the organization of scientific disciplines inherited the Cartesian body–mind dualism. An interdisciplinary approach is required to understand the necessary interface between memory traces (sensory experiences) and language (discursive connections ordering the objects of experience) for an individual and collective agency to take place at all. This challenge is taken by starting with the notion of modes of action, as defined in a previous work: "The entrepreneurial mode of action focuses on activities that produce objects exterior to oneself (*poiesis*) and strategies believed to be the most efficient to achieve one's goals (corresponding to Weber's "rationally-purposeful action"). The relational mode of action puts emphasis on relational configurations (it is close to Weber's *traditional social action* when it favours habits and routines that reproduce the social status and position of actors). The moral mode of action is based on the belief in the inherent worth of specific values (Weber's *value-rational action*). The identitary mode of action is based on the intersubjective definition of self (it partly corresponds to Weber's *affective social action* as drives also inform subjective identity). The motivational mode of action is the most complex one. It has no correspondence in Weber's typology of social action, it is closer to inquiry (Dewey, 1938)" [35] (p. 210). Symptomatically, it remains a blind spot in sociology as it was mainly left to psychology. One could say that affective social action is as close to the images of self as it is to the motivations: emotions are not only linked to drives but also to socially constructed objects and their varied meanings in different contexts. Global warming itself is objectified differently in different cultures, and hence, it motivates diverse types of reactions. Hence, modes of action offer a framework to approach the perceptual consciousness of global warming. As is shown, they allow specifying the notion of transactional horizons contributing to renewed understandings of agency.

## 3. Materials and Methods

The empirical material is constituted by 16 speeches delivered by Greta Thunberg in different rallies, with and without Extinction Rebellion, and in major events and con-

gresses such as, among others, the UN Climate Change Conference in Katovice, Poland (15 December 2018), the World Economic Forum in Davos, Switzerland (22 January 2019), the Goldene Kamera Film and TV Awards in Berlin (30 March 2019), the Houses of Parliament in London (23 April 2019), the French National Assembly in Paris (23 July 2019), the United States Congress in Washington (18 September 2019), and the UN General Assembly in New York (23 September 2019).

The choice to take Greta Thunberg as a case study guarantees that the data are not distorted in any way by the theoretical framework: there is no risk that the researcher influences the speeches of the only studied subject (Greta Thunberg) as they have neither met nor exchanged correspondence, and the data (speeches) are accessible worldwide so that anyone is in the potential position to critically assess the author's coding. Axial and selective codings [36] were applied. The iterative nature of grounded theory is central here. Reducing grounded theory to a linear and one-way inductive process starting with open coding is a misconception: a coding strategy is always informed by implicit or explicit representations, and hence, grounded theory can never be the outcome of pure induction. Its highly structured and optional aspects [37] must be recognized, as induction and deduction are recursively engaged in any analysis. Iterative procedures are involved in the methods used by scientists, but also in common sense (Section 5.2 comes back to this cycle in daily routines). The open, axial and selective codings cannot be separated *stricto sensu*, and acknowledgment of the iterative nature of coding is even the condition for the effectiveness and legitimacy of grounded theory. Open coding is never totally open, as existing social representations and/or scientific theories are already framing the questions that are asked to the material under scrutiny. Hence, it is necessary to situate the theoretical framework that is presiding over the ineluctable selection implied. The risk of blocking the induction–deduction cycles, which would derive from a top-down determination of the codes, can be limited only when the theoretical framework informs the coding strategy, yet does not determine the codes. It is important to insist on this fine nuance as it is safeguarding the iterative nature of coding and, hence, the relevance of grounded theory in its complexity. The theoretical framework has been presented (Section 2) to make sure that it only affects the coding strategy and not the codes themselves.

In the present analysis, the coding strategy is linked to the theoretical framework in the following way: considering speeches as sedimented representations of the social interactions that inform them, the coding procedure began with drawing attention to how these social interactions are reflected in the speeches of Greta Thunberg. The codes were first keywords and key expressions. This fostered awareness of the intentionality of discourse, and the connections made between the codes ended with the emergence of a central category that connects them all, namely "reflexive operations". Once this stage was reached, selective coding focused on the reflexive operations that are supposed to ground such a sedimentation of social interactions in the speeches of Greta Thunberg. Accordingly, axial coding was made by drawing connections between the codes. In axial coding, the codes were then expressed with verbs so as to situate the reflexive operations sustaining Greta's speeches. Therefore, the identified reflexive operations presided over the connections made between codes. These connections in turn produced new knowledge, as the iterative process recursively linking open, axial and selective coding entered the looping phase of repeated applications leading to the confirmation of the emergent theory. Along the process, some of the codes identified by Holmberg and Alvinius [3] and respective quotes were also integrated, and this demonstrates the all-embracing potential obtained once the process reaches the level of selective coding. The emerging knowledge, based on the findings (Section 4), is therefore both grounded on the theoretical framework (Section 2) and superseding it with a fine-tuned hypothesis (Section 5).

## 4. Findings

The logic of presentation is not the logic of discovery. Presentation of all the phases involved in the iterative logic of discovery (synthesized in Section 3) would be too long.

The latter involves a recursivity that is impossible to fully describe, and this is why the reader shall consider that the presentation made here can only be an imperfect account of the discovery. The presentation of the findings is therefore limited to the presentation of codes and respective quotes engaged in axial coding. These codes are presented in italics hereunder. In selective coding, the codes were then expressed with verbs (written in bold italic) so as to situate the reflexive operations sustaining Greta's speeches. These codes were finally attributed to the components (written in bold) of the theoretical framework (activities, relations, values, images of self, motivations) so as to see what new knowledge can be produced with it (Section 5).

It was found that, in Greta Thunberg's speeches, it was possible to attribute all codes and corresponding quotes to the different transactional horizons. This demonstrates that all modes of action are present in Greta's speeches. While some codes may fall under several transactional horizons and their corresponding modes of actions, they were attached to the most obviously concerned. As many quotes could illustrate the same codes, saturation of data attribution has been reached. Only the most illustrative quotes have been selected to keep the length of this paper within reasonable limits. The codes and quotes are presented hereunder, following the different transactional horizons concerned. For the quotes taken from the same source [9], only page numbers are indicated in parenthesis. For the quotes taken indirectly, from another source [3], the reference is indicated as in the original work.

**Entrepreneurial mode of action (Activities)**
*Denouncing wrongdoings:*
*The continued illusion of material production and development:*
"We live in strange world, where we think we can buy or build our way out of a crisis that has been created by buying and building things" (p. 40).
*Indulgence versus or resistance toward laissez-faire behavior [3]:*
"Neither of us ever mention the greenhouse gases, already locked in the system, nor that air pollution is hiding the warming so when we stop burning the fossil fuels we have already an extra level of warming, perhaps the size of 0,5 to 1,1 degree Celsius" [38].
"People keep doing what they do because the vast majority does not have a clue about actual consequences in our everyday life. They do not know that rapid change is required. We will think we know, we will think everybody knows but we don't. Because how could we?" [38].
*"Business as usual":*
"Dear Mr. Modi, you need to take action now against the climate crisis not just talking about it, because if you keep on going like this, doing business as usual and just talking about and bragging about the little victories. You are going to fail. And if you fail, you are going to be seen as one of the worst villains in human history, in the future. And you don't want that" [39].
*Silence of world leaders:*
"You would think that media and everyone of our leaders would be talking nothing else. But they never even mentioned it" [38].
*Praising good doings:*
"I dedicate this award to the people fighting to protect the Hambach forest. And to activists everywhere who are fighting to keep the fossil fuels in the ground" (p. 39).
**Relational mode of action (Relations)**
*Denouncing relations of domination:*
"We are about to sacrifice our civilization for the opportunity of a very small number of people to continue to make enormous amounts of money. We are about to sacrifice the biosphere so that rich people in countries like mine can live in luxury. But it is the sufferings of the many which pay for the luxuries of the few" (p. 13).
*Portraying intergenerational relations as a betrayal:*
"You lied to us. You gave us false hope. You told us that the future was something to look forward to" (p. 56–57).
*Stressing intergenerational responsibility:*

"What we do or do not do right now will affect my entire life, and lives of my children and grandchildren. What we do or do not do right now, me and my generation cannot undo in the future" [38].

**Moral mode of action (Values)**

*Affirming the superior interest of nature and civilization:*

"Our house is on fire" (p. 17).

"We are at a time in history where everyone with any insight of the climate crisis that threatens our civilization and the entire biosphere must speak out. In clear language. No matter how uncomfortable and unprofitable that may be" (p. 21).

*Affirming the need to change rules:*

"Today we use 100,000,000 barrels of oil every single day. There are no politics to change that. There are no rules to keep that oil in the ground. So we cannot save the world by playing by the rules, because rules have to be changed. Everything needs to change and it has to start today" [40].

*Setting aside selfish ways of living:*

"When I was about 8 years old, I first heard about something called climate change and global warming. Apparently that was something humans had created by our way of living" [38].

"Some people–some companies and some decision-makers in particular has known exactly what priceless values they are sacrificing to continue making unimaginable amounts of money. (...) I want to challenge those companies and those decision makers into real and bold climate action. To set their economic goals aside and to safeguard the future living conditions for humankind" [41].

*Sacrificing oneself:*

"We live in a strange world, where children must sacrifice their own education in order to protest against the destruction of their future" (p. 39).

**Identitary mode of action (images of self)**

*Presenting children as unheard victims:*

"And the saddest thing is that most children are not even aware of the fate that awaits us. We will not understand it until it's too late. And yet we are the lucky ones. Those who will be affected the hardest are already suffering the consequences. But their voices are unheard" (p. 57).

*Presenting children as climate activists:*

"And I agree with you, I'm too young to do this. We children shouldn't have to do this. But since almost no one is doing anything, and our very future is at risk, we feel like we have to continue" (p. 31).

*Presenting children as able and responsible agents:*

"So we have not come here to beg the world leaders to care for our future. They have ignored us in the past and they will ignore us again. [...] We have come here to let them know that change is coming whether they like it or not. The people will rise to the challenge. And since our leaders are behaving like children, we will have to take the responsibility they should have taken long ago" [40].

*Identifying with scientists:*

"There is one other argument that I can't do anything about. And that is the fact that I'm 'just a child and we shouldn't be listening to children'. But that is easily fixed–just start to listen to rock-solid science instead. Because if everyone listened to the scientists and the facts that I constantly refer to then no one would have to listen to me or any of the other hundreds of thousands of schoolchildren on strike for the climate across the world" (pp. 30–31).

"And just for quoting and acting on these numbers–the scientific facts–we receive unimaginable amounts of hate and threats" (p. 78).

*Presenting herself as a child with Asperger syndrome:*

"I have Asperger's syndrome, and to me, almost everything is black or white, I think in many ways we autistic are the normal ones and the rest of the people are pretty strange" (p. 6).

"Solving the climate crisis is the greatest and most complex challenge that *Homo sapiens* have ever faced. The main solution, however, is so simple that even a small child can understand it. We have to stop our emissions of greenhouse gases. And either we do that or we don't. You say nothing in life is black or white. But that is a lie. A very dangerous lie. Either we prevent a 1.5°C of warming or we don't. Either we avoid setting off that irreversible chain reaction beyond human control–or we don't. Either we choose to on as a civilization or we don't. That is as black and white as it gets" (p. 19).

*Presenting herself as a hated child:*

"Recently I've seen many rumours circulating about me and enormous amounts of hate" (p. 23).

"( . . . ) all the politicians that ridicule us on social media, and have named and shamed me so that people tell me that I'm retarded, a bitch and a terrorist, and many other things" (p. 3).

*Presenting herself as an independent child:*

"Some people mock me for my diagnosis. But Asperger is not a disease, it's a gift. People also say that since I have Asperger I couldn't possibly have put myself in this position. But that's exactly why I did this. Because if I would have been 'normal' and social I would have organized myself in an organization, or started an organization by myself. But since I am not good at socializing I did this instead" (p. 28)

**Motivational mode of action (motivations)**

*Looking for alternatives:*

"We live in a strange world, where no one dares to look beyond our current political systems even though it's clear that the answers we seek will not be found within the politics of today" (p. 40).

"And if solutions within this system are so impossible to find then maybe we should change the system itself?" (p. 14).

*Intensifying willingness:*

"We live in a strange world, where all the united science tells us that we are about eleven years away from setting off an irreversible chain reaction, way beyond human control, that will probably be the end of our civilization as we know it" (p. 39).

The recursive coding process revealed core reflexive operations that underpin Greta Thunberg's speeches. The codes and respective quotes show that all transactional horizons are involved in Greta's journey as a climate activist. With codes grouped in actions, it is possible to arrive at the level of the general reflexive operations that presides over Greta Thunberg's transactional horizons. Transactional horizons form a system that is actualized by reflexive operations: objectify activities, personify relations, sanctify values, unify images of self, diversify motivations. These reflexive operations are considered as vectors of agency, represented in Figure 2 (hereunder).

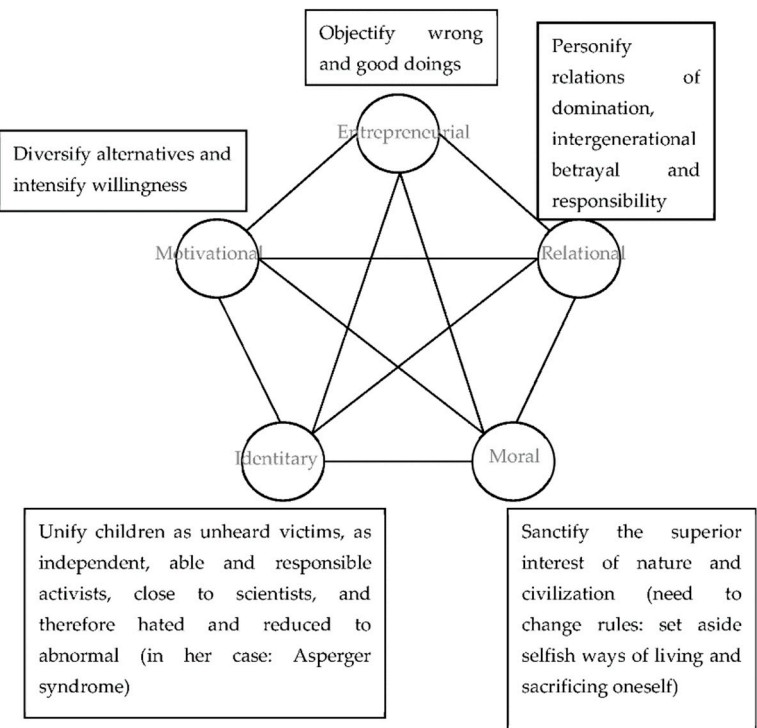

**Figure 2.** Vectors of agency of Greta Thunberg.

Figure 2 suggests that Greta's ways of acting are tightly knit together. In her speeches, Greta Thunberg makes many links among these different dimensions. The following quotes illustrate some of these links.

*Linking intergenerational relations, the value of knowledge and the motivation for strikes:*

"We are facing an existential crisis. The biggest crisis mankind have ever faced! Yet, it has been ignored for decades by those who knew about it. You know who you are, you who have ignored it, you are the most guilty! And it ain't us who stands here. We are young. We have not contributed to the crisis. We have just been born in this world and suddenly there was a crisis ahead of us that we are forced to live with. We, and our children and our grandchildren. And all future generations. We will not accept it. That's why we strike. We strike because we want a future and we will continue" [42].

*Linking relations and motivations:*

"(we live in a strange world ... ) Where a football game or a film gala gets more media attention than the biggest crisis humanity has ever faced. Where celebrities, film and pop stars who have stood against all injustices will not stand up for our environment and for climate justice because that would inflict on their right to fly around the world visiting their favourite restaurants, beaches and yoga retreats" (p. 41).

*Linking motivations and activities:*

"Some people say that I should study to become a climate scientist so that I can 'solve the climate crisis'. But the climate crisis has already been solved. We already have all the facts and solutions. All we have to do is to wake up and change. And why should I be studying for a future that soon will be no more, when no one is doing anything whatsoever to save that future? And what is the point of learning facts within the school system when the most important facts given by the finest science of that same school system clearly mean nothing to our politicians and our society?" (p. 10).

It is not necessary to present more quotes for this demonstration: not only are all transactional horizons present in Greta's speeches, but they are also tightly knit together. This has a very powerful effect, as shown by Greta Thunberg's speech that received the greatest media coverage, the one she delivered before the UN General Assembly, in New

York, on 23 September 2019: "This is all wrong. I shouldn't be standing here. I should be back in school on the other side of the ocean. Yet you all come to us young people for hope? How dare you! You have taken away my dreams and my childhood with your empty words. And yet I'm one of the lucky ones. People are suffering. People are dying. Entire ecosystems are collapsing. We are in the beginning of a mass extinction. And all you can talk about is money and fairy tales of eternal economic growth. How dare you!" (p. 96). Greta Thunberg tightly knits the aspects of climate change in just a few sentences. She shows the consequences of the entrepreneurial mode of acting ("money" and "fairy tales of eternal economic growth") on all other dimensions: relations among people and with nature ("you have taken away my dreams and my childhood with your empty words" "people are suffering", "ecosystems are collapsing"), images of self ("I should be back in school", "yet I'm one of the lucky ones"), values and motivations ("how dare you!"). It is the connection between the different dimensions of experience that makes her speech so powerful. The speech has influence not only because it takes place in a very prestigious place with a worldwide audience (many other speeches delivered at the UN fora actually had limited impact) but also because it conveys a vivid image of climate justice in a very concise speech. This concentration of all the aspects of climate justice in a vibrant speech invigorates Greta's emotional tone. It may be suggested that this emotional tone stems from the obligation to "say it all" in the short time allocated to Greta at the UN General Assembly. Further interdisciplinary research on emotions [43] would be needed here.

It appears that transactional horizons are central in the social negotiation for the definition of reality. They back Greta's constant call for the respect of science. While a scientific consensus about the increase of temperatures caused by greenhouse gases has been reached in recent decades, it has turned into a "political" controversy as it is challenged by non-scientific organizations or individuals, mainly because of corporate interests. The latter use the media and especially social media to throw doubt on the scientific consensus. Greta is well aware of this as she portrays power relations and underlines responsibility: "During the last months millions of schoolchildren have been school-striking for the climate and creating lots of attention for the climate crisis. But we children are not leaders. Nor are the scientists, unfortunately. But many of you here today are. Presidents, celebrities, politicians, CEOs and journalists. People listen to you. And therefore you have an enormous responsibility. And let's be honest. This is a responsibility that most of you have failed to take. You cannot rely on people reading between the lines or searching for the information themselves. To read through the latest IPCC report, track the Keeling Curve or keep tabs on the world's rapidly disappearing carbon budget. You have to explain that to us, repeatedly. No matter how uncomfortable or unprofitable that may be" [9] (pp. 69–70). The scientific consensus about global warming is encapsulated in a wider responsibility of explaining scientific facts, which Greta Thunberg depicts as "uncomfortable" and "unprofitable". Hence, it takes more than scientific evidence to convince people: the reference to objective reality is gaining attention when transactional horizons, triggering emotions, are conveyed.

It appears that the more transactional horizons are involved in one's discourse, the more chances it has to attract attention from a wider audience. It looks like knitting transactional horizons is attracting a larger audience than apologetic rhetoric displaying just one horizon and mode action. This can be exemplified by Greta's reference to Martin Luther King in her speech before the US Congress in Washington on 18 September 2019: "I also have a dream. That governments, political parties and corporations grasp the urgency of the climate and ecological crisis and come together despite their differences–as you would in an emergency–and take the measures required to safeguard the conditions for a dignified life for everybody on earth" (p. 85). This speech is not portraying a radical program for power redistribution. On the contrary, it calls for measures beneficial for "everybody on earth". Greta's dream is not that of a completely different world; she rather wishes a return to "normal life" for the youth: "Because then we millions of school-striking youth could go back to school" (p. 85). By not challenging the social positioning

of children, she is attracting sympathy from adults who would also like to see children "back in school". If marching children would not just question the generational order [11], but attempt to overthrow it, then the demographic basis of the climate justice movement might certainly shrink. By reassuring adults with regard to social positionings ("This is all wrong. I shouldn't be standing here. I should be back in school . . . "), Greta embraces a wide audience. This suggests that extremism is bound to shrink the use of transactional horizons to just one mode of action. The neo-liberal dogma, legitimizing the exploitation of natural resources, is one such extremism. The one that, precisely, leads to depletion of nature and consequently global warming. By seeing nature through all transactional horizons, Greta Thunberg is actually undoing this centration on only one mode of action (the entrepreneurial) and invites to consider the environment in relational, moral, identitary and motivational terms. The discussion below offers further insights into the dynamics of transactional horizons that contribute to renewed understandings not only of children's agency in the climate justice movement but power in general.

## 5. Discussion

A new approach to agency can be derived from the findings (Section 4): as core reflexive operations form a structure of transactional horizons, this implies that agency is exerted *through* structure, and not despite of or at the margins of structure. The Giddensian definition of structure as "a virtual order of transformative relations" [27] (p. 17) is used here to suggest that agency is what instantiates specific configurations of such an order. Agency is the fluid process giving structural properties to local arrangements. These structural properties are then presiding over the replicability of practices. The fact that climate marches are happening so often suggests that their structural properties are transposable in multiple contexts because they transport ecological concern through the transactional horizons that are most widely shared: activities, relations, values, images of self and motivations. This system formed by transactional horizons can contribute to the understanding of the two topics that are discussed hereunder, namely (Section 5.1) the exercise of agency through structure, and (Section 5.2) power as the naturalization of cultural pertinences.

### 5.1. The Exercise of Agency through Structure

In the year and a half between Greta's first school strike and the outbreak of the COVID-19 pandemic, climate marches took place regularly. These marches are expected to resume once deconfinement allows demonstrations again, and they might continue for as long as decision-makers do not take the appropriate measures to "save the planet". However, we still have to understand why climate marches happened so fast in different places. The rapid mobilization thanks to social media [3,4] explains the coordination within and communication among climate marches. However, it does not explain why, in the first instance, adhesion to Greta's concern was so large. The "starification" of the "lone girl" is not a convincing explanation: the media do not create events, they magnify them. The image of a girl sitting on the street and school-striking for climate was of course a trigger. Media coverage would have vanished, however, if Greta's deed did not rest on reasons that are shared widely enough to be able to lead to more school strikes and climate marches. Under the surface of events, there are deeper structures that preside over social movements. In order to highlight these deeper structures, the analytical perspective that is needed must be as detached as possible from the normative component of the phenomenon. The perspective must supersede the concern. Thus, in order to understand why climate marches are so often replicated, we must look at what makes a practice replicable at all.

The replicability of practices is approached here with the structuration theory: if reproduced social practices instantiate a "virtual order of transformative relations" [27] (p. 17), then this virtual order is logically the basis for the replicability of practices. In the structuration theory, this virtual order is defined as follows: "To say that structure is a 'virtual order' of transformative relations means that social systems, as reproduced

social practices, do not have 'structures' but rather exhibit 'structural properties' and that structure exists, as time-space presence, only in its instantiations in such practices and as memory traces orienting the conduct of knowledgeable human agents" [27] (p. 17). Seeing structure as a "virtual order of transformative relations" that is instantiated by social actors means that structure is constitutive of agency: agency is displayed *through* structure. By contrast, the perspective viewing agency *within* structure is reifying both terms as it rests on the assumption of a zero-sum game (what one loses is gained by the other). In this perspective, agency is considered as some "room for maneuver" left by the poor assembly of the parts of a construction. However, agency is what *instantiates* structural properties: the structure is the *means* for actors to instantiate some aspects of the whole structure. Hence, the structure is not a finished "whole" that contains or constrains some tiny part of "agency". On the contrary, it is a set of means for agency to instantiate structure in almost endless ways. Transactional horizons hence form a system of opportunities for social actors. Their concrete realizations may be scarce or extended, qualitatively poor or rich, depending on the articulation between the system of opportunities and the skills of actors. Greta's call for science is an instantiation of the structure: the constant reference to science is, in her view, a legitimation, while in the eyes of climate sceptics, science is manipulated or even, to some, a plot. The fact that science itself is disputed changes the system of opportunities: it no longer seems sufficient to refer to "the finest science" [9] (p. 10) as the contention about the objectivity of facts is overtly expressed even by some democratically elected leaders and as, consequently, the "reproduced social practices" that constitute social systems tend to include more beliefs and subjective views than empirically demonstrated truth.

The paradigmatic shift from "agency *within* structure" to "agency *through* structure" can therefore highlight how and to what extent social actors can instantiate structural properties according to the system formed by the "virtual order of transformative relations" [27] (p. 17) that is at hand in the contexts where they are acting. This means, more concretely, that human realizations are the outcomes of what actors seize in their surroundings to be transformed into opportunities. Eventually, reproduced social practices embody the most stabilized outcomes of this interplay. They sediment into institutions, which are provisionally stabilized forms, including concrete constructions. Consequently, there is no structure without agency. Structure exists through agency just as well as agency is built through structure. The "egg and chicken" thinking is irrelevant for social reality: structure and agency have no ontological existence; they are just *instantiated* in concrete realizations. Concrete realizations are conditioned by a structured language. Nevertheless, this language can be shared and reproduced only through the connection of the sensible and the intelligible, but the sensatory experience in social relations is much overlooked. Sociologists tend to forget the sensible experience as they emphasize the intellectual aspects of social bonding. Meanwhile, action needs to be understood both sensitively and intellectually. When this is the case, other social actors can "understand" a practice in the fullest way as the intelligible is connected to the sensible. The connection between reflective and perceptual consciousness is the basic mechanism that can explain the replications of practices (e.g., the marches for the climate). Nonetheless, it remains insufficiently understood, probably because the "disciplining" of sciences has split the study of phenomena into specialized and hardly reconcilable parts. Meanwhile, witnessing such huge marches, and especially being part of a movement made of multiple activities, with high levels of sensatory implication, leaves deep "memory traces". These massive and all-engaging activities are close to what Marcel Mauss [44] calls a "total social fact", shifting configurations in the "virtual order of transformative relations" [27] (p. 17).

Political controversies over science about global warming signal that opportunities that are recognized and seized by social actors are predominantly framed by the entrepreneurial mode of action (neo-liberal use of natural resources) that respects science only when it is profitable. In such a context, the objective evidence of global warming is not sufficiently respected to become a social total fact. Greta's deploring the disrespect for science makes her speak out, but she is a disputed candidate for this role: as a child, she embodies

the paradox of childhood in Western societies, depicted as "authentic" and at the same time "incompetent" [8,10–12]. This paradoxical social construction of childhood actually weakens her call for the respect of science in the eyes of climate skeptics, who do not hesitate to despise Greta Thunberg, such as Donald Trump avoiding her during the UN General Assembly (23 September 2019), or even to resort to attacks on her personality. This shows that despite scientific consensus over global warming, the status of science itself is politically polarized and that, consequently, the attacks on Greta Thunberg are instrumental: they use the socially constructed characteristics of children and childhood (naïve and incompetent) in Western societies to downplay what she says, hence affecting the overall reception of scientific evidence.

Nevertheless, as shown by the case of Greta Thunberg, children are able to counter domination. Hence, the political agency of children depends on and shapes the opportunities given by the configurations of transactional horizons. Agency and structure are not two separate entities, but the same thing in different states. After the reification of social structures [30], many sociologists have reified agency. A middle path can be found with a systemic approach that fosters renewed attention on the recursivity of social dynamics by considering that a virtual order (structure) and the transformation (agency) of its configurations are constitutive of each other. It is therefore irrelevant to oppose agency and structure, and even more misleading to consider agency at the individual level and structure at the collective level. Agency and structure are not the opposing terms of a binary; they are both dual notions as the one contains the other. This idea is also developed by others in their own terms: "Fuchs argues that rather than treat structure and agency as dual processes, they should be considered as variations along a continuum. Hence 'instead of persons and agency, sociology might start with variations in social structures'" (Fuchs, 2001:30). Emirbayer and Mische (1998) argue that structure and agency are mutable processes. Empirical social action is never completely structured (or determined) but neither is agency ever completely free of structure. The empirical challenge then becomes one of 'locating, comparing and predicting the relationship between different kinds of agentic processes and particularly structuring contexts of action' (Emirbayer and Mische, 1998:1005). This involves the recognition that actors' engaged responses to situational circumstances may initiate different forms of agency. The way forward according to Hitlin and Elder (2007) is to map out how different types of agency emerge in different situated contexts" [11] (p. 130).

Agency and structure are both observable at the level of concrete individuals. When observing individuals, what shall be put under scrutiny is the *duality* of agency, as a personal route with structural properties [27] (p. 17). The replicability of practices across contexts is strongly linked to what is able to sediment as memory traces: imitating, and hence reproducing, practices can only happen if these practices are understood not only intellectually, but also on a sensatory basis. The importance of the sensatory basis of personal experience highlights the controversies about the value of science. Children and young people engage in these controversies primarily through a figure with which they can identify (Greta). It is not some abstract epistemological debate that triggers them, but foremost a contemporary "hero" fighting against obscurantism for the survival of the species. This has great attraction power, as it reinforces simplified dichotomies (the good and the bad) and narratives that can more easily mobilize people to take sides. The distrust of European youth in the capacity of their respective governments to deal with global warming is highly correlated with expressive motivations: on 15 March 2019, a large survey in 9 European countries and 13 cities, where 1905 responses were gathered in a systematic random sample of protesters, showed that "45% of all school students agreed with the statement that Greta Thunberg had been a factor in their decision to join the Climate Strike" [45] (p. 4). While Greta relies on some complex scientific explanations that, for the sake of understanding, she translates in rather accessible terms in her speeches, these are not elements that are easily replicated in the arguments of climate marchers. Before the climate marches induced by Greta Thunberg, the awareness of pupils regarding the mechanisms

of climate change was low [46]. It looks like raising interest in the environment takes some charismatic aspects, and that, more generally, confidence versus distrust in science is mostly bound to who is speaking in the name of science or against it. This is illustrated by the fact that the "missed encounter" of Greta Thunberg and Donald Trump appears as a more important event for the media and the general public than the scientific facts that were discussed at the UN General Assembly. Media sensationalism is a very strong indicator of the importance of the sensatory perceptions in people's reflective consciousness.

This is precisely what "climate strikes" need in order to be replicated across countries: children can exert this form of political agency because they can both understand intellectually and feel sensitively what it means to be part of a group doing "climate strikes". It is a particular social movement, as it is not defending some identity recognition. It is not addressing the interests of just a portion of humanity; it is urging everyone to take measures for the survival of humanity itself. This is precisely what makes the case sociologically quite enlightening: the dynamics of climate strikes rest on structural properties that can be grasped at the level of individuals, as the latter *need* to incorporate them if they want to be heard. These properties are institutional opportunities, such as the ones given by science, the media, and their interplay. Nonetheless, these institutional opportunities are constructed in social negotiation implying transactional horizons and modes of action. The latter are enabling and constraining opportunities: these more sensitive categories (activities, relations, values, images of self and motivations) are structuring social practices that reproduce the orientation of institutions and actors. They form a symbolic structure mediating their relationships. This view is consistent with the structuration theory [27] and brings further insights into social dynamics: transactional horizons form a structure of symbolic categories that are used in the social practices enabling and constraining institutional opportunities. There is a twisting movement that is addressed now.

A transactional horizon can be defined as an interface between memory traces (sensory experiences) and language (discursive connections ordering the objects of experience). This translation is a complex process that transforms the simultaneity of senses into the hierarchy of discourses. This hierarchizing process rests on the physical law of utterances: every sentence takes time to be said: "sounds, in fact, can only be articulated one by one" [47] (p. 96; author's translation). This physical law of utterances provokes a hiatus between thought and language: "Language cannot represent thought, from the start, as a whole; it has to arrange it part by part in a linear order" (Ibid. *author's translation*). Consequently, the hierarchized structure of language, coming from the physical law of utterances, is conducive to the ordering of things. As we must say things one after the other, they are organized into dominant and complementary themes. The social ordering of things follows context-specific prioritizations. Transactional horizons form a system that is actualized by reflexive operations: objectify activities, personify relations, sanctify values, unify images of self, diversify motivations. These reflexive operations (forms of being) are at the interface between memory traces and language. It is therefore necessary to focus on the central unit of human language: verbs. Narratives are built through the designation of objects and their relations connected by verbs. Michel Foucault considers that human language is rooted in the verb *to be*: "The whole essence of language is collected in this singular word" [48] (p. 109). Foucault adds: "The entire species of the verb is reduced to the one that means: *to be*. All the other verbs secretly use this unique function, but they have covered it with determinations that hide it ( . . . )". (Ibid.). These determinations come from added attributes regarded as inherent to the subjects: for instance, "she writes" refers to "her, as a writing person". Hence, every verb designates the potentiality of *being in the capacity of . . .* , which are actually *forms of being*. Core reflexive capacities are linked to sensory perceptions that can be reverted in core expressions: objectifying (produce concrete forms), personifying (give a personal nature to forms), sanctifying (declare forms as holy), unifying (stabilize forms), diversifying (explore variations of forms).

A new definition of agency can be derived from these systemic vectors of agency: agency is being in the capacity of intervening on things through objectification, personifica-

tion, sanctification, unification and diversification. The vectors of agency are derived from the subject–object relation whereby the action of the subject on the object (transitive verb) includes the capacity to symbolically transform the object. This new definition of agency is more precise than "the capacity to make a difference" which does not indicate how this capacity is exerted. Retrospectively, it also highlights the same absence of the "how" in current definitions of agency, mainly based on choice, such as in Giddens' definition: "Agency concerns events of which an individual is the perpetrator, in the sense that the individual could, at any phase in a given sequence of conduct, have acted differently. Whatever happened would not have happened if that individual had not intervened. Action is a continuous process, a flow, in which the reflexive monitoring which the individual maintains is fundamental to the control of the body that actors ordinarily sustain throughout their day-to-day lives" [27] (p. 9). Saying that "agency concerns events of which an individual is the perpetrator" leaves intact the problem of the actions that are perpetrated, as they are innumerous. Any action can create an event. Adding that the individual can act differently does not resolve the problem, as this only points to the possibility of choosing another action, hence provoking another event. In the end, the problem remains because acts are thought of as if they would be separate. In reality, it is not possible to isolate acts [33]. Moreover, action is not just a sequence of acts, it also implies the coordination of the senses at each stage of the microprocesses involved. To take just an example, a simple conversation is made of so many microprocesses that it is impossible to fully detail them. Therefore, isolation of acts is currently made because we merely focus on the apparent actions. If agency cannot be reduced to apparent actions, as it is concerned with the systemic dynamics of transactional horizons, then we have to ask what is orienting this symbolic transformation of objects. This is where power comes in.

### 5.2. Power as the Naturalization of Cultural Pertinences

Equipped with a rather sophisticated oral language that seems to open myriads of possible symbolic constructions, human beings are nevertheless limited by the physical law of utterances when it comes to sharing these constructions. The latter, consequently, also tends to naturalize cultural preferences. If the physical law of utterances plays a major role, then the narratives that tend to become dominant are the ones to which more time and energy are dedicated. The time of speech allowed to different persons, according to their status and to the locations where their views are taken into account, is an indicator of power distribution. Greta Thunberg's time to speak has been dramatically raised in the process leading from her first school strike to the platform given to her before the UN General Assembly in New-York (23 September 2019). Nonetheless, she had to "say it all" in one still relatively short time allocated to her. It may be suggested that her emotional tone on this occasion stems from the power imbalance between the time and media coverage dedicated to scientific evidence on one side and the denial of global warming through irresponsible jokes behind air-conditioners on the other side. Greta's emotions probably surfaced also because of this deep and hard-felt injustice, all the more intriguing when denial of scientific evidence comes from leaders of nations that have developed mainly thanks to sciences and technologies. Greta's emotional tone created simultaneously more sympathy and more antipathy, but her emotional tone is already and outcome of power imbalance. Hence, the widening gap between partisans and opponents of climate justice is a recursive movement stemming, ironically, from denial of sensible evidence. It is therefore a great irony when climate skeptics accuse Greta Thunberg of being a divisive person. The analysis made along the perspective of transactional horizons shows that Greta Thunberg reflects about nature with all transactional horizons. By widening consideration for the environment in relational, moral, identitary and motivational terms, Greta Thunberg is actually challenging the centration on the entrepreneurial mode of action that is reducing nature to exploitable resources. This may entail reduction in power imbalance and corresponding profits, and this is what is felt as threatening by many. The growing attention to what Greta Thunberg

says comes from her ability to use all transactional horizons and not reduce the environment to just one.

From the findings, a relevant hypothesis emerges here, namely that the use of all transactional horizons is what allows Greta Thunberg to widen the audience and consequently also the time allocated to her speeches. This hypothesis is consistent with the fact that domination of an ideology can only be opposed with the multiplication of points of view. It is a relevant hypothesis with regard to social movements as reactions to insufficiently acknowledged diversity. The findings (Section 4) lead to renewed considerations on inquiry [31] as an instrument for agency. In order to understand agency and its link to structure, or the continuity between the organization of individual experience and social organization [29], we have to better understand the reflexive operations involved in inquiry. Two major considerations can be briefly made here. The first one is that social actors develop their daily inquiries in ways that are similar to what scientists do, and the second one is that inductive and deductive reflexive operations made on a daily basis by social actors to make sense of their environment imply social transactions. Everybody resorts to these two recursive movements that are called induction (reasoning from particular to general) and deduction (reasoning from general to particular). The only difference between scientists and laypersons is the degree of sophistication of their methods, but everyone is engaged in a logic of inquiry [31] where indication and extrication of objects are central: "Anything of which a human being is conscious is something which he is indicating to himself ( . . . ). ( . . . ) to indicate something is to extricate it from its setting, to hold it apart, to give it a meaning or, in Mead's language, to make it into an object". [19] (p. 20). The operations of indication and extrication can be specified as in Figure 3 (hereunder).

**THEORY**

**Deduction**

| | |
|---|---|
| Replacing the objects of focus in the general picture of representations and theories (using language) | Using theories and representations to focus on specific objects (using language) |
| Indicating to oneself the kind of objects in focus by naming them (using concepts) | Naming the objects in focus (using concepts) |
| Specifying the extricated objects by comparing them to one another (attributing characteristics) | Confirming the known characteristics of the objects (attributing characteristics) |
| Extricating objects from a background (isolating shapes) | Exploring the unknown characteristics of the objects (isolating shapes) |
| Perceiving the environment through sensatory faculties (relying on five senses) | Opening the perception of the objects to all senses (relying on five senses). |

**Induction**

**DATA**

**Figure 3.** Inductive and deductive operations.

As Figure 3 suggests, inductive and deductive operations have inverse relationships to hierarchy: induction starts from the less hierarchized elements (data), while deduction starts from the most hierarchized level (theory). Induction follows a route of progressive conflict resolution leading to the most hierarchized structure that is language (any definition of a word embodies the resolution of conflicting meanings). Reversely, deduction imposes a hierarchized view on reality, but reality always resists language to some extent and hence obliges one to make sense of yet unknown characteristics. This is where perceptions are eventually opened to new reassessments of reality leading to new progressive conflict resolutions. Hence, every person is continuously engaged in induction–deduction cycles, striving for balanced conduct (assimilation and accommodation in Piagetian terms) [49].

The emergent theory of transactional horizons, grounded on the reflexive operations at play in Greta Thunberg's speeches, assumes that inductive and deductive operations (hence balance movements) are marked by social transactions. Especially the heart of inductive and deductive operations, the central stage of the process (see Figure 3), is most permeable to social transactions: when specifying the extricated objects and confirming their characteristics, social actors are in fact negotiating the attribution of the "event" or "phenomenon" under scrutiny to a specific "transactional horizon". When an object can be attributed to all transactional horizons, or "read" through all these perspectives, it becomes a central issue to which more time is dedicated as people fight around its definitions and their consequences. This is the case of the "environment". It may become a *total social fact* [44] when viewed through all transactional horizons (activities, relations, values, images of self and motivations). All claims and counterclaims are then negotiated in the social transactions made through these transactional horizons. The climate justice movement is able to reshape identities because nature is becoming part of the *Self* more and more [28]. This identitary expansion is not yet fully understood, and its sociological importance is overlooked by reductionist media focus on the "lone girl". Greta Thunberg is an oxymoron: a child with political agency. The mediatic centration on her personality is blinding us, like a tree hiding the forest. What we need to understand is that the polarized projections made on Greta Thunberg are instantiations of the controversies over scientific truth. As has been suggested, scientific evidence is sustained by sensitive categories: the reference to objective truth is mediated by cultural preferences. They are, in the end, enabling and constraining institutional opportunities: science, the referential grounding of Greta Thunberg's speeches, is received in different ways according to institutional configurations. The latter are the dynamic outcomes of negotiations that are made on a daily basis by social actors using sensible categories that are shaping their modes of action. The legitimacy of science does not solely rest on the rigor of objective demonstrations but also on the concrete people making them known.

A new vision of agency is appearing with the hypothesis that the use of all transactional horizons is what allows Greta Thunberg to widen the audience and consequently also the time allocated to her speeches. Her agency is deployed thanks to the connections she makes between activities (broadly speaking: "save the planet") and their relational, moral, identitary and motivational aspects. This contributes to a new paradigm—agency *through* structure—considering the continuum between fluid reflexive operations (agency) and their sedimented outcomes (structure). This emerging paradigm therefore looks at the recursive relations between agency and structure thought of as two different states of reflexive operations. The transactional horizons of Greta Thunberg illustrate this continuum: by looking at the reflexive operations underpinning her speeches, the analysis locates the co-determination of agency and structure. The widening audience of Greta Thunberg is therefore more than the effect of media coverage focusing mainly on her entrepreneurial capacity: the social movement of climate justice is fostered through the structure of transactional horizons that are all conveyed in Greta's speeches. The paradigm of agency *within* structure sees Greta's success as an individual achievement constrained by limiting forces, reinforcing the iconic image of the hero. The new paradigm of agency *through* structure supersedes this perspective by showing that the media "success story" of Greta Thunberg

is itself an outcome of the domination of the entrepreneurial mode of action, which was actually implicit in the foundations of the "New paradigm in childhood sociology" 30 years ago: "Children are and must be seen as active in the construction and determination of their own social lives" [50] (p. 8). The narrative of the "competent child" stems from the domination of the entrepreneurial mode of action. The alternative perspective of agency *through* structure arising here is therefore taking distance from this regime of truth [1] by considering that agency is deployed through the structure of transactional horizons and that, consequently, they are both outcomes and means of social action, an insight deriving from structuration theory [27]. Consequently, transactional horizons are the "virtual order of transformative relations" [27] (p. 17), and their recursivity explains why "agency *through* structure" is a movement going in both directions, from fluid states (agency) to sedimented outcomes (structure) and reversely. This new vision retrospectively highlights some blind spots in the agency *within* structure perspective. First, this narrative rests on the same entrepreneurial mode of action that has been naturalized by the globalization of the market economy, whereby agency signals the competence of the actor within structural constraints. This should foster a critical look at "childhood agency" as the core element in this concept is "the idea that children can be seen as independent social actors" [51] (p. 3). Second, this narrative of the competent actor reinforces the Cartesian divide between body and mind, whereby agency (mind) would free oneself from the contingencies of structure (body). Thus, it is losing touch with the material basis of social life as, typically, senses are still left out of consideration in the sociology of childhood, merely because of disciplinary boundaries.

The physical law of utterances implies that the sensory and the intelligible faculties of human beings are bound to a twisting movement that turns the simultaneity of perceptual consciousness (senses) into a hierarchized reflective consciousness (language). Accordingly, the perspective of agency exerted *through* structure opens a view of agency lying in the ability to translate experience in diversified social transactions, and reversely to enrich experience with complex configurations of transactional horizons. Agency and structure are two states on a continuum of fluid and sedimented forms, reflecting the interface between perceptual and reflective consciousness. The emergent theory of transactional horizons situates this interface in the human ability to connect objects with verbs. The core reflexive operations of objectification, personification, sanctification, unification and diversification are central vectors of agency. They have a structure that must be seen in continuity with institutional arrangements. Accordingly, agency and structure are just the concepts we use to designate the recursive movements on a continuum stretching from fluid to sedimented states: social structures are sedimented agency and agency is social structure in progress. Agency and structure are the same thing, but in two different states. There is no dualism (dichotomies between separate entities, inherited from the Cartesian body-mind divide), there is just human duality (two simultaneous forms of being, bodily and spiritual existence). Reflective consciousness tends to hierarchize the duality (considering one expression of being as superior to the other). It is this tendency that must be more fully understood if we want to become more reflexive on our own determinations.

The social process turning Greta Thunberg into a "star" confirms the relevance of this emergent theory: the mediatic discourse exploiting the oxymoron of a "lone girl" having political agency is an outcome of the twisting movement translating simultaneous perceptions of the environmental change into a hierarchized language about climate change. Media sensationalism twists Greta's endeavor into the dominant entrepreneurial regime of truth [1]. Nevertheless, this is a peculiar instantiation of a general mechanism linking perceptual and reflective consciousness. Twisting movements qualify beings who are able to imagine causes that are not directly visible. Hence, this specific characteristic of human beings [52] makes them able to envisage the causes of global warming, only indirectly experienced through the perturbations linked to it (floods, droughts, etc.). This is precisely what also makes them able to hierarchize claims in conventional ways, through social contracts and conventions such as human rights instruments, the Paris agreement

and similar engagements. Only humans can believe in such abstract reconstructions of reality and fight to undoing and redo these conventionalized frames. Inquiry [31] is about reducing the discrepancy between beliefs and reality. Being overconfident in limited frames of understanding has always put humans in conflict with the material reality. Global warming signals that accommodation is urgent. The scientific evidence of the human causes for global warming is disputed by the proponents of an entrepreneurial mode of action which they would like to see freed as much as possible from state control. The extraordinary agency of Greta Thunberg is probably a sign that the claim for the recognition of other modes of action is the real movement occurring beneath the surface of climate marches. More space for other modes of action would transform power relations. The question is, again, whether the common good will prevail over personal interests.

## 6. Limits and Future Directions

The limits of this paper are bound to the interdisciplinary endeavor needed to better understand the links between cognitive and political factors in the interplay of perceptual and reflective consciousness. The dynamics of the three "instantiations" of reflexive operations in Mead's theory of identity (I, Me, Self) should be addressed, because Mead's "objective relativism" [33] (p. 5) gives insights into how objects are constructed through symbolic interactions modifying the components of identity. The intuition about identity expansion with a global self-inclusivity of nature is probably a relevant path to follow, as understanding nature and culture as two "complexly interwoven ecological systems" [53] (p. 4) is still ahead of social scientists. Greta's accounts on activities, relations, values, images of self and motivations are clear enough, at least in Western cultures, to allow a proper understanding of what is at stake in her speeches. Greta's speeches embody a presentation of self and discursive strategies that are understandable in those terms. Further research would be needed to situate the limits of the public reception of Greta's speeches in non-Western societies. This would potentially be a major contribution to a more reflexive "ecological turn" that humanity needs to make in order to recover its adaptation capacity and eventually heighten its survival chances.

With a view of the transformation of the sensory world into a hierarchized language based on a twisting movement, it looks like power, in the end, is bound to voice. The attitude of former US President Donald Trump denying the scientific evidence of global warming incorporates this political distortion of perceptual consciousness. His contemptuous avoidance of Greta Thunberg when he passed her in the halls of the UN General Assembly on 23 September 2019 was one of the most telling instantiations of a worldwide habitus [54] that consists in obeying those who speak louder. More interdisciplinary research is needed to address this twisting movement that is naturalizing power imbalances. This could be achieved through a dialogue with recent approaches on emotions [42], while also going back to psychosocial classics on perceptual and reflective consciousness [28,33] as they contain some gems that might have been forgotten. Political sciences would also contribute to unfold the power relations in this twisting movement and hence further highlight the mechanisms explaining how cultural preferences are naturalized. By including the physical law of utterances as a central element constraining discourses, social sciences could reconcile with cognitive sciences, biology and physics and, maybe, highlight how much political extremism is linked to the reduction of the range of transactional horizons and corresponding modes of action. The neo-liberal legitimation of environment exploitation is paradoxical with the recognition on the common dependence on limited natural resources. Exhaustion of resources may also, more specifically, be the outcome of the globalized dominance of the entrepreneurial mode of action. This makes the study of climate justice movements all the more important.

## 7. Conclusions

A theory of transactional horizons emerges from the empirical findings made through the analysis of Greta Thunberg's speeches. The polarized reception of her call for the respect of science is linked to the symbolic structure of transactional horizons that is mediating the relationships between social actors, science and the media. These relations of power are exacerbated by Greta's speeches. Climate strikes can therefore be considered as instantiations of larger cultural preferences and institutional arrangements. They trigger renewed engagement for a cause (save the planet) that may actually foster a potentially important redistribution of power. These findings contribute to the structuration theory [27] with a focus on the pragmatic categories used in daily life (activities, relations, values, images of self and motivations) and the corresponding modes of action (entrepreneurial, relational, moral, identitary and motivational) that are constitutive of societal orientations. The findings presented and discussed in this paper lead to the hypothesis that Greta Thunberg's extraordinary agency is bound to the use of all transactional horizons in her speeches, as this widens her audience and hence the potential replications of demonstrations supporting her goal, namely climate justice. Further research is needed to test this hypothesis on a large scale. A growing understanding of perceptual and reflective consciousness as a recursive process might have great social impact. Greta Thunberg's rise is a symptom of this perceptual change; it is not a purely "social" or "political" construction. The material basis of this movement, the link between global warming and perceptual consciousness, must be better understood as, in turn, this knowledge about the material basis of reflective consciousness is needed to accommodate human behavior to the environment. The perspective of the *Anthropocene* [55] could therefore benefit from a better understanding of recursive movements in the physical reactions to human overconfidence in its own rationality. Eventually, heightened consciousness of recursivity may contribute to policies acting accordingly and maybe avoid the ineluctability of continuous global warming.

**Funding:** This research received no external funding.

**Data Availability Statement:** All data can be found in the material published by Greta Thunberg and listed in the references.

**Conflicts of Interest:** The author declares no conflict of interest.

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
