# Peer review of "The Transactional Horizons of Greta Thunberg"

_societies, doi:10.3390/soc11020036_

Round 1
Reviewer 1 Report
I see this manuscript for the second time, and I wish to state again that it seems to be very perfect. The author(s) improved it greatly and responded to all my queries. So, I find this paper fully acceptable.
Reviewer 2 Report
El trabajo, bien argumentado, presenta una multitud compleja de conceptos interdisciplinarios, que permitirían realizar unos cuantos capítulos.
Agrupados, en este trabajo, resultan algo difíciles de asimilar para el lector que no esté súper especializado en ellos.
Una síntesis algo más clara, que evite el exceso literario, que ayudara al lector a no perderse, sería deseable, planteando la teoría de los horizontes transaccionales, tal como propone el autor, como una comprensión de la estructura de los discursos.
En definitiva, una articulación más clara entre el problema, la metodología y los resultados, que lleve a las conclusiones sin perderse en la compleja estructura conceptual propuesta.
The work, well argued, presents a complex multitude of interdisciplinary concepts, which would allow a few chapters to be carried out.
Grouped together, in this work, they are somewhat difficult to assimilate for the reader who is not super specialized in them.
A somewhat clearer synthesis, which avoids literary excess, which helps the reader not to get lost, would be desirable, proposing the theory of transactional horizons, as proposed by the author, as an understanding of the structure of discourses.
In short, a clearer articulation between the problem, the methodology and the results, leading to conclusions without getting lost in the complex conceptual structure proposed.
Reviewer 3 Report
Evaluating this paper overall academic value is extremely difficult. There are parts which are truly brilliant and others that seem repetitive, inconsistent and even unhelpful. The lack of focus is sometimes troublesome.
The original idea of the paper to propose a new analysis of Greta Thunberg's speeches is interesting and deserves attention.
However the elaboration of the theoretical framework seems to be the real goal of the essay while Greta Thunberg's speeches analysis becomes only a pretext for such philosophical developments - that are at times very powerful and instructive but also at times rather confusing.
To our knowledge the pages 17 to 26 could be cut without harming so much the whole thing. The long passage on music and tonality can be acceptable in a book chapter of a monography but seems really "off" in terms of academic papers.
We will suggest the author to devote more time to the content of the speeches, to their scientific basis, to go back a little to the issue at stake - that is: environmental awareness, environmental concerns and climate crisis.
One of the dimension that seems to lack in the analysis is the reference to reality as the most powerful force backing up Greta's speeches. Not only the moral, relational, emotional and social aspects but also simply the referential aspect. The call for science, for taking into account scientific discourses is not exactly the same as the call for traditions or established practices. One aspect of the reading and reaction to Greta's discourses is linked to the problem of legitimacy of science - to which she refers - as becoming in itself an object of dispute and contention. Some epistemological background on scientific controversies will be a pretty important addition to make.
Otherwise the discussion about agency and structure is rather convincing.
But many references are too cryptic for the general reader. What is exactly the "ontological turn" of children studies? Others are too deep to be but broached up in this way. For example, the reference to Descola sounds very exciting but we are just left at it without any real stuff to substantiate it: "The hypothesis about the success of Greta Thunberg depicting the extraordinary agency of a child in the climate justice movement can be further refined by putting this movement in a broader perspective, namely of the four cosmologies (naturalism, animism, totemism and analogism) identified by Descola [55] and their influence on discourses about climate change." Here the real question: can we really simply see in environmental contemporary discourses a new entanglement of these four anthropological modes of being? Indeed the problem is that they are normally mutually exclusive in Descola's ontology. Or is not a kind of overcoming of naturalism? And thus a fifth category slowly coming into being. But now, as for as this paper is concerned, the issue will be to ground that idea on Greta's speeches, demonstrate it to the reader rather than contemplating heuristic hypothesis.
I am hesitating between rejection and serious rewriting.
Round 2
Reviewer 3 Report
The author has taken into consideration my previous remarks and provided a new paper that is substantially revised and improved. It gains clarity and focus as well as demonstrates a capacity to engage with critical comments in a productive and efficient way. Though the author's analysis of Greta Thunberg's speeches may still be seen more as a mean to assert a theoretically-based methodological framework than a real case study of Greta Thunberg's speeches social and political underpinnings, it provides a revised understanding of the relations between agency and structure that is very helpful. One thing perhaps could have discussed in conclusion is replicability. The "exceptionnalization" and "starization" of Greta Thunberg: is it something positive that will inspire others or is it a way to make it unique and thus not replicable? The crux of the agency/structure relation is replication & dissemination - in this way individual agency becomes really social. In another paper, the author may discuss this issue. A comparative perspective with similar Greta Thunberg figures in different political regimes (for example China) will be interesting.
This manuscript is a resubmission of an earlier submission. The following is a list of the peer review reports and author responses from that submission.
Round 1
Reviewer 1 Report
In this manuscript the author tried using the theoretical framework of transactional horizons to analyze the 16 speeches from Greta Thunberg. This manuscript had a strong front end and connected a wide breadth of related literature. Having said that, I have several major concerns for this manuscript.
First, the author has done a great job in introducing how transactional horizons channeled social interactions. But, how were these social interactions reflected in Greta Thunberg’s speeches? What I see is that the author classified various quotes into each horizon while the social interaction parts were generally missing. The theoretical framework was not tightly intertwined with data used in this study.
Second, the author put a lot of effort in discussing the duality between agency and structure. This theory and related research questions should be investigated with social movement, or more specifically, the climate march data. Again, I feel the great disconnection between the theoretical framework and the data used in this study.
Third, the author mentioned that axial and selective coding strategies were utilized in the analysis of the 16 speeches from Greta Thunberg. How many people conducted the coding work separately? In case of disagreement, was there an independent third person to make a judgement? Was the codebook available to readers to replicate the analysis?
There were some minor issues.
(1) The author used both “Holmberg & Alvinius” and “Holmberg and Alvinius” and should stick to one of them.
(2) The author emphasized the pragmatic questions such as “what do you do?”, “who do you know?”, “what do you think?”, “who are you?”, “what do you want?” in lines 226-227. I expect the author to situate the speeches from Greta Thunberg into these questions. Otherwise, what was point to bring them up?
(3) The sentence in lines 235-237 had grammatical error.
Reviewer 2 Report
In my opinion, this is somewhat unusual and one of the most interesting contributions I have ever reviewed. Undoubtedly, it is up-to-date in all aspects. It provides with novel insights not only in the childhood studies, but also in the understanding of the modern environmental activism. The manuscript is organized logically and well-written. The cited literature is relevant, and its amount in enough. I find it very suitable to 'Societies'. After several rounds of critical reading, I see only a few issues for small amendments.
- 'Society' is a multidisciplinary journal, which means the readers may or may not be aware of some specific terminology. This is why I ask you to give very brief definitions of agency in both Abstract and Introduction. Additionally, I ask the authors to simplify Abstract to make it 100% clear also to the unprepared readers.
- I think the section 2 can be split into two sections, namely 2. Theoretical Framework and 3. Materials and methods (do not forget about subsequent re-numbering of the following sections).
- Lines 548-550: please, delete this redundant paragraph.
- I encourage (but do not insist) you to think and to explain whether the phenomenon of Greta Thunberg is possible and so bold in the only cultural frames and norms of the 'Western' societies.
- In two places, you use the term 'Anthropocene'. Well, but, please, cite the basic publications where it was proposed (for instance, the works by Crutzen and Zalasiewicz). Note, please, another word 'Anthropogene' circulated since the beginning of the 20th century thanks to the Russian scientists, but it was not so politicized as the Anthropocene.
- I'm sorry, but I can't understand why the figures are not given directly in the text.
- Please, try to avoid too short, single-sentence paragraphs where possible.
Good luck with revisions! I very hope to see your nice paper published soon!
Reviewer 3 Report
The main concept “transactional horizons” is not sufficiently grounded in previous literature. The aim of the article is not clear enough.
The discussion is too brief, the conclusion is long and unclear.
How does one person's media success prove the validity of a scientific theory? (1163) Isn't argumentum ad populum a fallacy after all?